# Revealing the Active Site of Gold Nanoparticles for the Peroxidase-Like Activity: The Determination of Surface Accessibility

**Ching-Ping Liu \*, Kuan-Chung Chen, Ching-Feng Su, Po-Yen Yu and Po-Wei Lee**

Department of Chemistry, Fu Jen Catholic University, New Taipei City 24205, Taiwan;
itick2230@gmail.com (K.-C.C.); sawada821228@gmail.com (C.-F.S.); leay3529@gmail.com (P.-Y.Y.);
jack991004@gmail.com (P.-W.L.)
**\*** Correspondence: 129723@mail.fju.edu.tw; Tel.: +886-2-2905-3575

**Abstract:** Despite the fact that the enzyme-like activities of nanozymes (i.e., nanomaterial-based artificial enzymes) are highly associated with their surface properties, little is known about the catalytic active sites. Here, we used the sulfide ion ($S^{2-}$)-induced inhibition of peroxidase-like activity to explore active sites of gold nanoparticles (AuNPs). The inhibition mechanism was based on the interaction with Au(I) to form $Au_2S$, implying that the Au(I) might be the active site of AuNPs for the peroxidase-like activity. X-ray photoelectron spectroscopy (XPS) analysis showed that the content of Au(I) on the surface of AuNPs significantly decreased after the addition of $S^{2-}$, which might be contributed to the more covalent Au–S bond in the formation of $Au_2S$. Importantly, the variations of Au(I) with and without the addition of $S^{2-}$ for different surface-capped AuNPs were in good accordance with their corresponding peroxidase-like activities. These results confirmed that the accessible Au(I) on the surface was the main requisite for the peroxidase-like activity of AuNPs for the first time. In addition, the use of $S^{2-}$ could assist to determine available active sites for different surface modified AuNPs. This work not only provides a new method to evaluate the surface accessibility of colloidal AuNPs but also gains insight on the design of efficient AuNP-based peroxidase mimics.

**Keywords:** gold nanoparticles; peroxidase-like activity; active site; surface accessibility

---

## 1. Introduction

Owing to their potential biological importance, the development of nanomaterials as enzyme mimics has received a great of interest recently [1,2]. Compared to natural enzymes, nanomaterial-based artificial enzymes (i.e., nanozymes) exhibit several advantages, such as low cost, easy preparation, high stability and robustness. In addition, metal, metal oxide and carbon-based nanomaterials possess several kinds of enzyme-like activities and have been applied in biosensing and immunoassays [3] as well as biomedical applications [4]. Despite the rapid progress achieved with nanozymes in terms of development and application, the relatively low catalytic efficiency of nanozymes still greatly limits further practical applications as compared to natural enzymes.

Enzyme-like activities are intrinsic properties of nanozymes, but they can be regulated by the manipulation of aspects such as the size, shape, morphology, surface modification, and composition [1,2]. Among these, size and surface coatings might be more dominant because catalytic reactions typically take place on the surface of nanozymes. It is straightforward to expect that small nanozymes possess relatively high catalytic activities due to the high surface-to-volume ratio. As for surface modification, it presents a dilemma between the colloidal stability and catalytic activities of the nanozymes [5]. Surface modification is indispensable in preventing the aggregation of colloidal

nanomaterials, which might decrease the catalytic performance by shielding or passivating surface active sites. For instance, unmodified gold nanoparticles (unmodified AuNPs, i.e., preparation of gold nanoparticles without additional surface stabilizers) with high gold-atom exposure on the surface displayed significantly higher peroxidase-like activity compared to those with low gold-atom exposure, such as cysteamine-capped AuNPs (Cys-AuNPs) [6]. This implied that the occupation of the gold surface by cysteamine could reduce the catalytic activity. In addition, several modification strategies such as changing surface charge, redox potential, acidity, and stability could also be utilized to modulate the catalytic activities of nanozymes [5]. To date, previous studies have explored the influences of surface properties on catalytic activities [1,2,5], but the active sites of enzyme-like activities are still not fully understood, and the systematic assessment on the surface ligands or stabilizers has yet to be achieved. Accordingly, researchers urgently need a fundamental approach to evaluate the surface accessibility manipulated by surface modification and to retain the optimal enzyme-like activities of nanozymes.

To address the above issue, it is essential to identify the active sites of the enzyme-like activities. In one study, the peroxidase-like activity of unmodified AuNPs could be inhibited by sulfide ions (i.e., $S^{2-}$) due to the interaction with Au(I) to form $Au_2S$ on the surface of AuNPs [7]. Similarly, halide ions (i.e., $Br^-$ and $I^-$, denoted as X) could also switch off the peroxidase-like activity of protein-modified AuNPs based on Au–X interactions [8]. The above inhibition mechanism was mainly attributed to surface passivation, which inspired our investigation of the active site for the peroxidase-like activity of AuNPs. In this work, we used the $S^{2-}$-induced inhibition of the peroxidase-like activity to explore active sites of gold nanoparticles (AuNPs). AuNPs decorated with common stabilizers such as citrate, cysteamine, polyvinylpyrrolidone (PVP) and gum Arabic (GA) [6,9,10] were chosen in this study and AuNPs prepared with $HAuCl_4$ and $NaBH_4$ only (i.e., the unmodified AuNPs) were also included for comparison. Since the peroxidase-like activities of AuNPs with various surface coatings could all be inhibited by $S^{2-}$, the surface Au(I) might be the active site of the peroxidase-like activity of AuNPs. The X-ray photoelectron spectroscopy (XPS) measurements showed that the content of Au(I) on the surface of AuNPs significantly decreased after the addition of $S^{2-}$, which might be contributed to the more covalent Au–S bond in the formation of $Au_2S$. In addition, the peroxidase-like activities of AuNPs with different coatings were compared by the oxidation of peroxidase substrates and the production of hydroxyl radicals (·OH), respectively, which were further confirmed by kinetic analysis. Importantly, the variations of Au(I) without and with the addition of $S^{2-}$ for different surface capped AuNPs were in good accordance with their corresponding peroxidase-like activities. Taken together, the XPS and kinetic results supported that the Au(I) on the surface of AuNPs was the active site of the peroxidase-like activity and the quantitation of accessible Au(I) could be responsible for the peroxidase-like activity of AuNPs.

## 2. Results

### 2.1. Characterization of AuNPs

Unmodified, citrate-stabilized AuNPs (Cit-AuNPs), cysteamine-capped AuNPs (Cys-AuNPs), and AuNPs coated with PVP (PVP-AuNPs) were chosen in this study because they have been reported to possess peroxidase-like activities [6,11,12]. In addition, AuNPs modified with GA (GA-AuNPs) were also included due to their great stability in physiological conditions [10]. Since the catalytic activity of AuNPs was size-dependent [13–15] and mainly contributed by the content of Au rather than those surface coating molecules, the size of all kinds of AuNPs should be carefully controlled during the preparation. Figure 1 showed transmission electron microscopy (TEM) images of different surface modified AuNPs followed by calculation of the size distribution. Roughly, the average size of all AuNPs was about 9.0 nm, and the difference between the largest unmodified AuNPs (9.4 ± 1.5 nm) and the smallest PVP-AuNPs (9.0 ± 1.6 nm) was just 0.4 nm, which was within the error of one standard derivation. In addition, UV-Vis absorption spectra of all AuNPs also showed similar features

in their surface plasmon resonance (SPR) bands (as shown in Figure S1). Therefore, these five kinds of AuNPs could be considered as equivalent in size (i.e., excluding all surface capping molecules). However, considering the slight difference in the size distribution of these AuNPs, the contribution to the peroxidase-like activity might not be totally ruled out. Importantly, previous work has reported that surface area has relatively little influence on the surface reactivity of AuNPs for the catalytic degradation of 4-nitrophenol while the size was distributed from 6.6 to 9.3 nm [16]. This suggested that the size control of our AuNPs (about 9.0 nm) was within the limited interference from surface area and suitable to study the correlation between the catalytic activity and surface capping molecules.

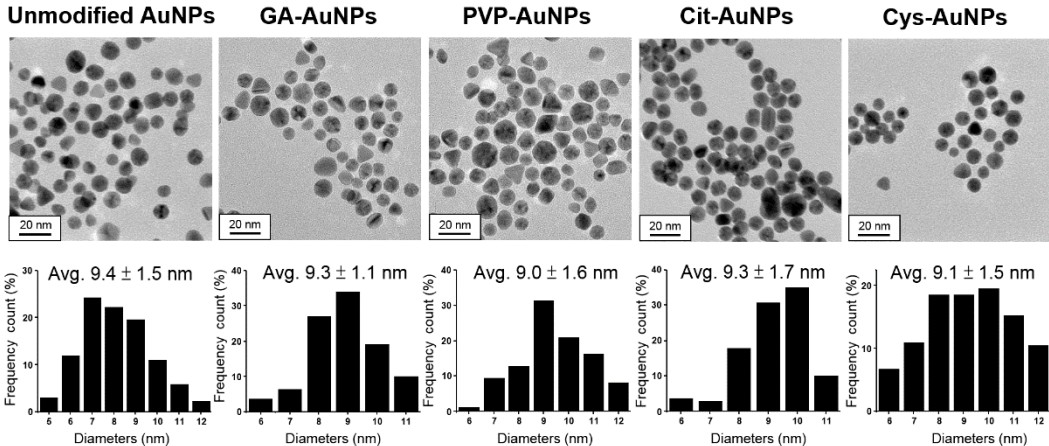

**Figure 1.** TEM images and size distributions of unmodified AuNPs, GA-AuNPs, PVP-AuNPs, Cit-AuNPs, and Cys-AuNPs, respectively.

### 2.2. XPS Measurements of AuNPs without and with $S^{2-}$ Addition

The surface passivation of AuNPs to inhibit the peroxidase-like activity by either sulfide ($S^{2-}$) or iodine ions ($I^-$) has been ascribed to the Au–S and Au–I bond-induced blocking of active sites, respectively [7,8]. In addition, $S^{2-}$ can react only with Au(I) and not with Au(III) to form $Au_2S$ in aqueous solution or at room temperature [17]. Accordingly, we hypothesized that Au(I) on the surface of AuNPs might be the active site of the peroxidase-like activity. Next, the inhibition of the peroxidase-like activity for different surface-capped AuNPs was examined by addition of $S^{2-}$. As shown in Figure 2a, the five kinds of AuNPs could catalyze hydrogen peroxide ($H_2O_2$) to oxidize the substrate of 3,3′,5,5′-tetramethylbenzidine (TMB), resulting in the appearance of the blue color. The control experiment without AuNPs was not observed any color change. This indicated that the five kinds of AuNPs all possessed the intrinsic peroxidase-like activities. When pre-mixing these AuNPs with $S^{2-}$, their peroxidase-like activities were significantly inhibited and thus the solutions retained the original red color of AuNPs (see Figure 2b). These results implied that AuNPs with various surface coatings should have the same active site of Au(I), and thus their peroxidase-like activities could all be inhibited by $S^{2-}$. It should be noted that excess $S^{2-}$ can also lead to aggregation of AuNPs [18,19], and thus the active sites may be reduced by increasing the size of AuNPs. To avoid this problem, the added amount of $S^{2-}$ was carefully controlled to inhibit nearly 90% of the peroxidase-like activity (see Figure S2). It meant that the surface passivation of these AuNPs should not be at a saturation level of $S^{2-}$. Following the previous work that $I^-$ could block active sites of AuNPs [8], our TEM images and UV-Vis spectra also verified that the inhibition of the peroxidase-like activity by $S^{2-}$ was due to blockage of the active sites on the surface rather than aggregation of AuNPs (see Figure S3).

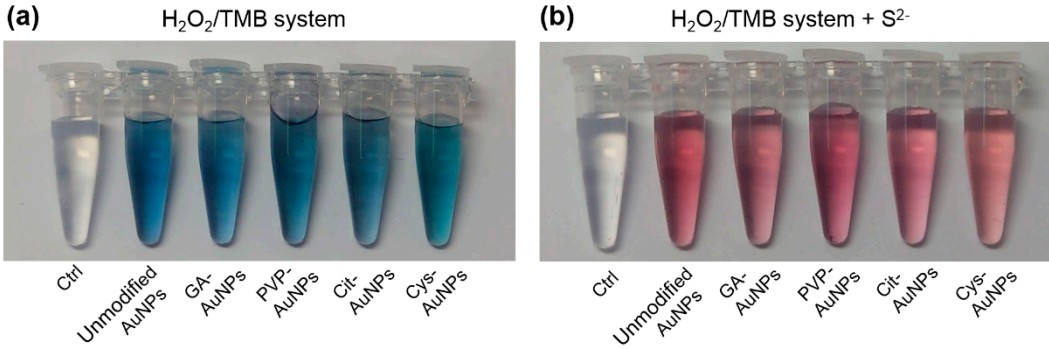

**Figure 2.** Experiments were carried out using all AuNPs with the same Au content in acetate buffer (10 mM, pH 4.0) and $H_2O_2$ (0.2 M) with TMB (0.5 mM) as the substrate. (**a**) In the $H_2O_2$/TMB system, all AuNPs showed the peroxidase-like activities; the color of TMB changed from colorless to blue. The same solution without AuNPs was as the control. (**b**) When premixing all kinds of AuNPs with $S^{2-}$, the peroxidase-like activities of AuNPs were all inhibited and thus the solutions retained the original red color of the AuNPs.

The $S^{2-}$-induced inhibition of the peroxidase-like activity of AuNPs is represented in Figure 3a. Accordingly, X-ray photoelectron spectroscopy (XPS) was employed to determine the Au(I) and Au(0) oxidation states before and after interaction with $S^{2-}$. Figure 3b shows the XPS spectrum with the two main peaks of Au(0) $4f_{5/2}$ and $4f_{7/2}$, respectively, and the peak of $4f_{7/2}$ was corrected to the binding energy at 84.0 eV as the reference [20].The two bands of Au 4f could be deconvoluted into two distinct components, which were attributed to Au(0) and Au(I) (see Figure 3b red and blue traces) [20–22]. In addition, the percentage of surface Au(I) was calculated based on the deconvoluted peak area of Au(I) and Au(0). Moreover, the change of the surface Au(I) was also determined based on the analysis of XPS spectra without and with $S^{2-}$ addition (Figure S4). As shown in Figure 3c, the percentage of Au(I) was significantly decreased when Au(I) interacted with $S^{2-}$ to produce $Au_2S$ [7]. It was speculated that the Au–S bond was more covalent than ionic in $Au_2S$ [23] and thus the surface Au(I) content should be reduced. Table 1 listed the calculated Au(I)% without and with the addition of $S^{2-}$, respectively, and $\Delta$Au(I)% (i.e., the change of Au(I)%). Among them, Au(I)% without $S^{2-}$ was represented the intrinsic Au(I)% on the surface of AuNPs, which was manipulated by different surface modification. The Au(I)% with $S^{2-}$ was only decreased part of the intrinsic Au(I)%, implying that the Au(I) on the surface of AuNPs was not fully accessible for the catalytic reaction. Of note, the added amount of $S^{2-}$ was just inhibited nearly 90% of the peroxidase-like activity and could not induce aggregation of AuNPs as stated above. Accordingly, the $\Delta$Au(I)% could be recognized as the accessible percentage of Au(I) for the peroxidase-like reaction. On the basis of Figure 3c and Table 1, the $\Delta$Au(I)% values from large to small were unmodified AuNPs > GA-AuNPs > PVP-AuNPs > Cit-AuNPs > Cys-AuNPs, which might contribute to their peroxidase-like activities (see the following text). The XPS spectra of S 2p after the addition of $S^{2-}$ were also taken (Figure S5), but bands of S 2p were very weak due to trace amount of $S^{2-}$ bound onto the surface of AuNPs and the intrinsic poor X-ray sensitivity factor of S as compared to Au [20]. By comparison, the analysis of the XPS spectra of Au 4f was relatively reliable. These XPS analysis indicated that complete inhibition of the peroxidase-like activity by $S^{2-}$ resulted in significant loss of surface Au(I). Therefore, we further confirmed that accessible Au(I) on the surface was the main requisite for the peroxidase-like activity of AuNPs.

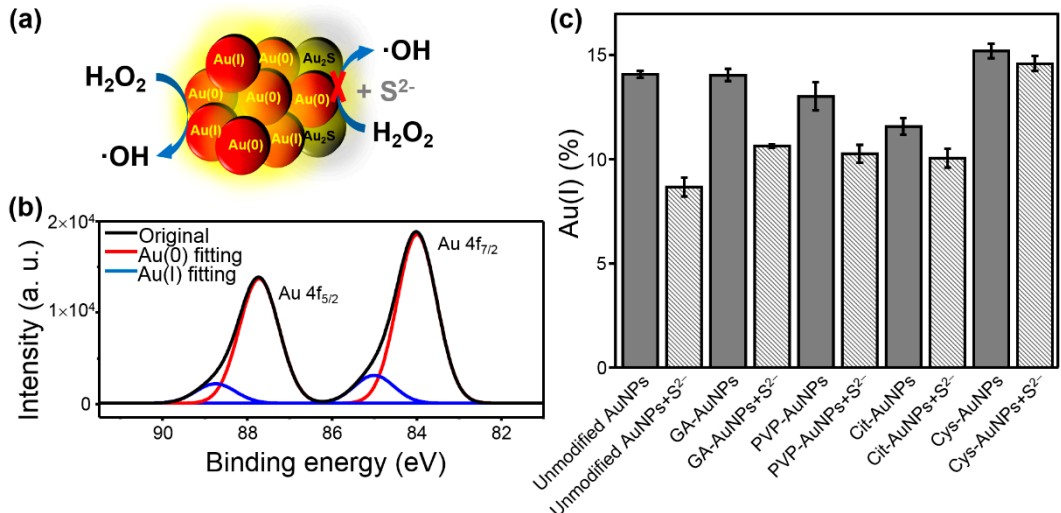

**Figure 3.** (**a**) The XPS spectra were deconvoluted into two distinct components, which were attributed to Au(0) and Au(I) as red and blue traces, respectively. (**b**) The change of surface Au(I)% without (gray bars) and with (stripped bars) addition of $S^{2-}$. (**c**) The change of surface Au(I)% without (gray bars) and with (stripped bars) addition of $S^{2-}$.

**Table 1.** XPS analysis of the surface Au(I) content without and with the addition of $S^{2-}$ for AuNPs with different surface modification.

| Sample | Au(I)% [1] Without $S^{2-}$ | Au(I)% [1] With $S^{2-}$ | ΔAu(I)% [2] |
|---|---|---|---|
| Unmodified AuNPs | 14.1 ± 0.2 | 8.7 ± 0.5 | 5.4 |
| GA-AuNPs | 14.0 ± 0.3 | 10.6 ± 0.1 | 3.4 |
| PVP-AuNPs | 13.0 ± 0.7 | 10.3 ± 0.4 | 2.7 |
| Cit-AuNPs | 11.6 ± 0.4 | 10.1 ± 0.5 | 1.5 |
| Cys-AuNPs | 15.2 ± 0.3 | 14.6 ± 0.4 | 0.6 |

[1] The content of surface Au(I) was expressed as a percentage of the total peak area of the Au $4f_{7/2}$ (peak area of Au(I) + peak area of Au(0)). [2] Δ Au(I)% = Au(I)% (without $S^{2-}$) − Au(I)% (with $S^{2-}$).

## 2.3. Comparison of the Peroxidase-Like Activity of Different Surface-Capped AuNPs

Once the Au(I) was the active site of AuNPs, the quantification of Au(I) could be a possible contributing factor for the corresponding peroxidase-like activity. To compare the peroxidase-like activities of different surface-capped AuNPs, all AuNPs were used with the equivalent Au content, which was either quantified by inductively-coupled plasma mass spectrometer (ICP Mass) or controlled with the same O.D. value (i.e., optical density for spectral absorbance) of their SPR bands. As shown in Figure 4a,b, unmodified AuNPs exhibited the highest catalytic activity among them, which is reasonable due to their surface might only stabilize by $BH_4^-$ and $Cl^-$ rather than any other modifiers [24]. In comparison, other AuNPs decorated with additional protectors or stabilizers could block or shield their active sites, thereby decreasing the catalytic activities. The arrangement of the peroxidase-like activity from high to low was as follows: unmodified AuNPs > GA-AuNPs > PVP-AuNPs > Cit-AuNPs > Cys-AuNPs. Of note, the peroxidase-like activity might be influenced by the charge characteristics between AuNPs and peroxidase substrates [6]. To test this possibility, the positively-charged peroxidase substrate, TMB, was replaced with negatively-charged ABTS (2,2′-azinobis(3-ethylbenzthiazoline-6-sulfonic acid)diammonium salt). With ABTS as the substrate (see Figure S6), the order of the peroxidase-like activity for Cit-AuNPs and Cys-AuNPs was exchanged, which was consistent with the previous report [6]. In fact, all but the Cys-AuNPs were negatively-charged AuNPs in acidic solutions [6,10,25]. For this reason, Cys-AuNPs possessed stronger affinity to ABTS than to TMB, leading to better catalytic activity than that of Cit-AuNPs

(see Figure S6). Therefore, other alternatives were needed to compare the peroxidase-like activities of all five kinds of AuNPs.

The possible mechanism of the peroxidase-like activity stemmed from the ability to decompose $H_2O_2$ into hydroxyl radicals (·OH) [26]. Since ·OH was the key intermediate for the oxidation of peroxidase substrates, the production of ·OH could be used to evaluate the peroxidase-like activity of these AuNPs. Terephthalic acid (TA) can react with ·OH to form TAOH, which has unique fluorescence at 435 nm, and thus it can be used as a fluorescent probe to monitor the ·OH generation [27]. The emission spectra of $H_2O_2$ mixed with different AuNPs were presented in Figure 4c. Obviously, unmodified AuNPs exhibited the most intense fluorescence among them (see Figure 4d), indicating that unmodified AuNPs possessed the best peroxidase-like activity. Importantly, the ranking of the ability to convert $H_2O_2$ into ·OH was in good accordance with the above measurements of the catalytic activity based on TMB oxidation (Figure 4a,b).

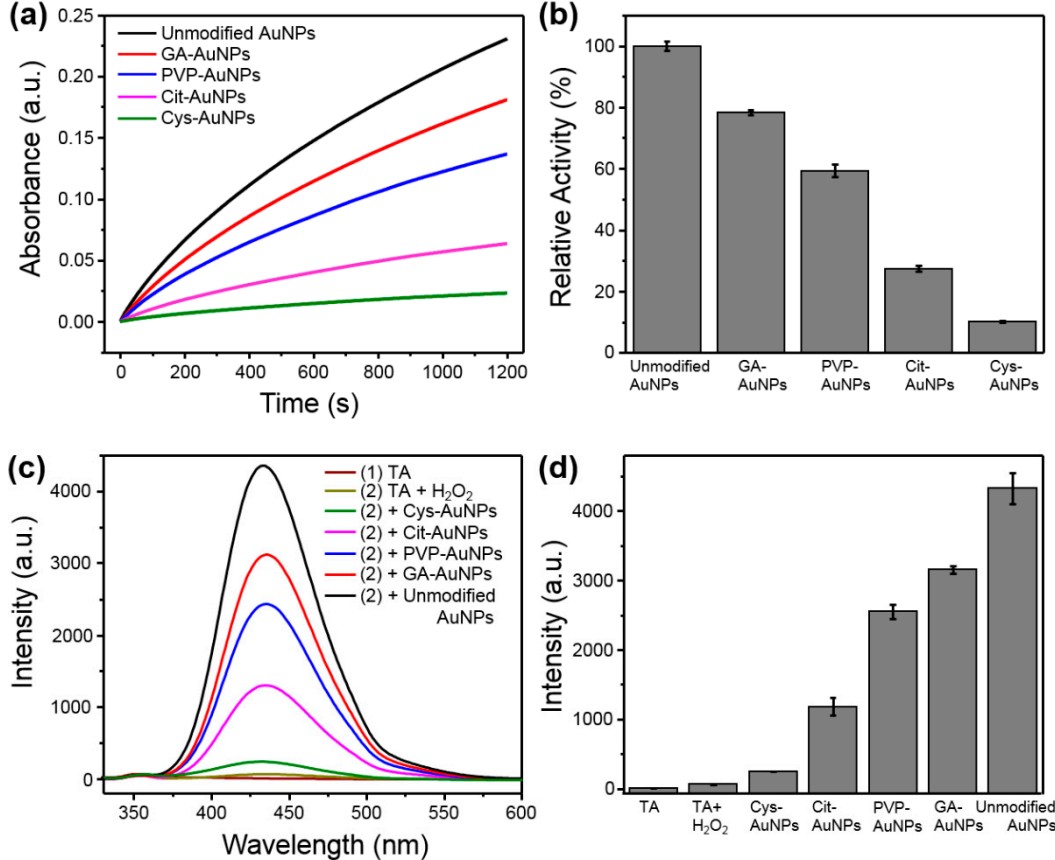

**Figure 4.** Comparison of the peroxidase-like activities of different surface modified AuNPs (with equivalent content of AuNPs at 1.0 O.D. cm$^{-1}$). (**a**) The time dependent absorbance at 652 nm of oxidized TMB. (**b**) The absorbance at 652 nm after the catalytic reaction with different AuNPs for 10 min. (**c**) The emission spectra of TAOH fluorescence intensities corresponding to the level of ·OH formed during the catalytic reaction in the presence of each kind of AuNPs. (**d**) The emission intensity at 435 nm after the catalytic reaction with different AuNPs for 10 min.

Further, the Michaelis–Menten model was used to analyze and compare the peroxidase-like activities of all five kinds of AuNPs. The kinetic parameters, including the Michaelis–Menten constant ($K_m$) and the maximum initial velocity ($V_{max}$), were obtained from Lineweaver–Burk plots as shown in Figures S7–S11 and are listed in Table 2 for comparison. The catalytic constant $k_{cat}$ is the turnover number per enzyme molecule and is equal to $V_{max}/[E]_0$, where $[E]_0$ is the initial concentration of enzyme. Since the $[E]_0$ for all AuNPs are equivalent as stated above, the $V_{max}$ values could be regarded as the catalytic constant $k_{cat}$. Obviously, the $V_{max}$ values decreased in the order of unmodified AuNPs >

GA-AuNPs > PVP-AuNPs > Cit-AuNPs > Cys-AuNPs, which was consistent with the aforementioned comparison of the peroxidase-like activities for all five kinds of AuNPs (Figure 4). Otherwise, the $K_m$ values were represented the binding affinity between enzymes and substrates, but no significant trend was obtained. This implied that the contribution from the electrostatic attraction between oppositely charged AuNPs and peroxidase substrates was relatively minor.

　　　Taken together, the $\Delta Au(I)\%$ values decreased in the order of unmodified AuNPs > GA-AuNPs > PVP-AuNPs > Cit-AuNPs > Cys-AuNPs, which were in good accordance with the arrangement of their corresponding peroxidase-like activities (Figure 4 and Table 2). This indicated that the calculation of $\Delta Au(I)\%$ could represent available active sites for the peroxidase-like activity of AuNPs. Therefore, the addition of $S^{2-}$ could assist to determine the surface accessibility of different surface modified AuNPs, which would benefit the quantitation of other gold-based catalysis.

**Table 2.** Comparison of the kinetic parameters for the peroxidase-like activities of different surface modified AuNPs.

| Catalyst | Substrate | $K_m$ (mM) | $V_{max}$ ($\times 10^{-3}$ mM s$^{-1}$) |
|---|---|---|---|
| Unmodified AuNPs | $H_2O_2$ | 113 ± 1 | 4.26 ± 0.05 |
| GA-AuNPs | $H_2O_2$ | 84 ± 1 | 2.23 ± 0.01 |
| PVP-AuNPs | $H_2O_2$ | 96 ± 1 | 1.91 ± 0.01 |
| Cit-AuNPs | $H_2O_2$ | 151 ± 1 | 1.05 ± 0.01 |
| Cys-AuNPs | $H_2O_2$ | 213 ± 1 | 0.310 ± 0.002 |
| Unmodified AuNPs | TMB | 0.1880 ± 0.0005 | 4.17 ± 0.03 |
| GA-AuNPs | TMB | 0.0942 ± 0.0002 | 1.82 ± 0.01 |
| PVP-AuNPs | TMB | 0.0551 ± 0.0004 | 1.22 ± 0.02 |
| Cit-AuNPs | TMB | 0.0493 ± 0.0007 | 0.527 ± 0.002 |
| Cys-AuNPs | TMB | 0.0528 ± 0.0001 | 0.118 ± 0.001 |

## 3. Discussion

　　　According to the XPS analysis, to determine the content of accessible Au(I) for all five kinds of AuNPs, the effects of their corresponding surface capping molecules could be potentially evaluated. Typically, unmodified AuNPs without additional surface modifiers could retain the highest number of active sites on the surface, resulting in the best catalytic performance. Since PVP and GA are typically used to protect colloidal AuNPs without any chemical bonding in-between, the surface accessibility of PVP-AuNPs and GA-AuNPs might be confined by the steric hindrance of such macromolecules (i.e., PVP and GA) around the surface of AuNPs [28]. As for Cit-AuNPs, the possible explanation was that a fraction of the surface gold atoms was oxidized to the Au(I) state either by binding with citrate $COO^-$ or by balancing the adsorption stability of negatively-charged citrate [20]; thereby, the accessible Au(I) on the surface of Cit-AuNPs might also be restricted. It should be noted that Cys-AuNPs exhibited relatively large contents of intrinsic Au(I) (see Figure 3c), but their peroxidase-like activity was still poor. In the literature, thiols are well known to poison the surface of AuNPs (i.e., catalytic deactivation) due to the formation of strong Au–S bonds [29,30]. Cysteamine with the –SH group could oxidize superficial gold atoms (i.e., Au(0)) after binding to AuNP surface, resulting in a significant contribution to inaccessible Au(I) because such Au(I)–thiolate complexes are inactive catalytically [31]. Accordingly, the Au(0) and Au(I) on the surface are both indispensable for the peroxidase-like activity of AuNPs. Since the use of $S^{2-}$ addition only specifically reacted with Au(I) to inhibit catalytic activity, definitive confirmation of Au(0) as the active site was challenging in this work. To determine Au(0) on the surface of AuNPs by thiols, further experiments are under way.

　　　Mechanistic studies have reported that silver nanoparticles (AgNPs) decomposed $H_2O_2$ to produce ·OH through a Fenton-like reaction involving the oxidation of Ag(0) to Ag(I) ($\varphi = -0.7996$ V) and the reduction of $H_2O_2$ ($\varphi = +1.77$ V) [32]. Since the peroxidase-like activity of AuNPs was highly associated with the accessibility of surface Au(I), the mechanism might be similar to that of AgNPs. However, the oxidation of Au(0) to Au(I) ($\varphi = -1.83$ V) was not favorable in the presence of $H_2O_2$. In

addition, the SPR band of AuNPs was almost unchanged during degradation of $H_2O_2$ (see Figure S12), indicating the absence of conversion from Au(0) to Au(I). This observation was consistent with the previous work, which considered AuNPs as a catalyst rather than a Fenton-like reactant in the presence of $H_2O_2$ [11]. Further studies will be needed to explore the mechanism of the peroxidase-like activity and the role of the Au(I) as the active site on AuNPs.

## 4. Materials and Methods

### 4.1. Chemicals

Polyvinylpyrrolidone (PVP), hydrogen tetrachloroaurate(III) tetrahydrate ($HAuCl_4 \cdot 3H_2O$), sodium citrate, gum arabic, sodium borohydride ($NaBH_4$), hydrogen peroxide ($H_2O_2$, 30%), 3,3′,5,5′-tetramethylbenzidine (TMB), 2,2′-azinobis(3-ethylbenzthiazoline-6-sulfonic acid) diammonium salt (ABTS), sodium sulfide nonanhydrate ($Na_2S \cdot 9H_2O$), ascorbic acid (AA), 2-aminoethanethiol (cysteamine), terephthalic acid (TA), and horseradish peroxidase (HRP) were purchased from Taiwan Sigma-Aldrich (Merk KGaA, Gernsheim, Germany). All reagents were used without further purification.

### 4.2. Preparation of Different Surface-Capped AuNPs

Unmodified AuNPs were prepared according to a previous report [6]. Briefly, $HAuCl_4$ solution (500 µL, 25 mM) was diluted with D.I. water (39.5 mL), and then $NaBH_4$ solution (1 mL, 1%) was slowly injected under vigorous stirring (the injection rate was about 200 µL min$^{-1}$). After the solution changed color from pale yellow to red, the resulting solution was continuously stirred in the dark for 1 h.

The preparation of PVP-AuNPs followed a seed-mediated growth method that allows easy control of the size of AuNPs [9]. First, 555 mg of PVP (molecular weight 10 kDa) was added into an aqueous solution of $HAuCl_4$ (1 mM, 50 mL) and the mixture was stirred at 4 °C for 30 min. Second, 5 mL $NaBH_4$ (100 mM) was rapidly added into the mixture under vigorous stirring. After stirring at 4 °C for another 30 min, the resulting solution was kept as seeds for preparation of PVP-AuNPs. Next, 1 mL $HAuCl_4$ (50 mM) was diluted by D.I. water (35 mL), 555 mg PVP was added, and the solution was stirred at 4 °C for 30 min. Then 10 mL seed solution was mixed with the above solution and stirred for another 30 min. Finally, 15 mL AA (5 mM) was slowly dropped into the above mixture and the solution stirred at 4 °C for 2 h.

GA-AuNPs were synthesized in aqueous solution by using the preparation method of PVP-AuNPs with slight modifications. The only difference was that 555 mg of PVP was replaced by 333 mg of gum Arabic (molecular weight 58 kDa). The other steps were all the same as those in the preparation protocol of PVP-AuNPs stated above.

Citrate-capped AuNPs were synthesized according to a previously published method [6]. Initially, $HAuCl_4$ solution (1 mL, 25 mM) was dissolved in D.I. water (99 mL) and heated up to boiling. Next, sodium citrate solution (3 mL, 1 wt%) was quickly added to the boiling solution. The color changed from pale yellow to red, indicating the formation of Cit-AuNPs.

Cysteamine-capped AuNPs (i.e., Cys-AuNPs) were synthesized by following a previous publication with some modifications [6]. Briefly, $HAuCl_4$ solution (500 µL, 25 mM) was diluted with water (39.5 mL), and then cysteamine solution (85 µL, 213 mM) was added to the $HAuCl_4$ solution. After stirring for 20 min at room temperature, 2.1 mL $NaBH_4$ solution (10 mM) was added and the mixture was stirred for more than 10 h in the dark.

All glassware and stir bars were cleaned in aqua regia (1:3 $HNO_3$:HCl) and rinsed with D.I. water before use for the preparation of AuNPs. Except for Cys-AuNPs, other kinds of AuNPs were given after centrifugation at 10,000 rpm to remove excess reactants and washed with D.I. water at least three times. All final solutions were stored at 4 °C in a refrigerator for further experiments.

### 4.3. Characterization

Transmission electron microscopy (TEM) images were taken with a JEM2100 FEG (JEOL, Hitachi, Japan) transmission electron microscope at an accelerating voltage of 200 kV. UV-Vis absorption spectra of various AuNPs were obtained by a JASCO V-670 (JASCO Corp., Tokyo, Japan) spectrophotometer. Emission spectra were recorded by a JASCO FP-8300 fluorescence spectrophotometer. High resolution X-ray photoelectron spectroscopy (HRXPS, PHI Quantera SXM, ULVAC-PHI Inc, Osaka, Japan) was used to determine atomic compositions. Deconvolution of XPS peaks and calculation of peak area were performed with the peak-fitting function in Origin 9.0 software package (OriginLab Inc., Northampton, MA, USA). The Au contents of the different surface modified AuNPs were measured with an inductively-coupled plasma mass spectrometer (ICP Mass, Agilent 7500ce, Agilent Technologies Inc., Santa Clara, CA, USA).

### 4.4. Examination of the Peroxidase-Like Activity of AuNPs

The peroxidase-like activities of all AuNPs were evaluated at pH 4 (acetate buffer, 10 mM) with either TMB (0.5 mM) or ABTS (0.5 mM) as peroxidase substrates in the presence of 0.2 M $H_2O_2$. HRP was used as the positive control. For the inhibition of peroxidase-like activity, 10 μM $S^{2-}$ was first mixed with each kind of AuNPs (1.0 O.D. cm$^{-1}$) for 10 min. Subsequently, the acetate buffer, $H_2O_2$ and TMB were added. After the solution incubated for 30 min, color change of the resulting solution was obtained with the spectrophotometer (JASCO V-670) and photographs were taken.

To monitor the catalytic oxidation of the peroxidase substrate with time, absorbance values were collected at 420 nm for ABTS or at 652 nm for TMB using the scanning kinetic mode of the UV-Vis spectrophotometer.

### 4.5. Detection of Hydroxyl Radicals during the Peroxidase Reaction

Terephthalic acid (TA) was used as a fluorescent probe ($\lambda_{ex}$ = 315 nm/$\lambda_{em}$ = 425) nm to detect ·OH. Initially, TA was dissolved in ethanol (0.6 mM). Every solution of AuNPs (1.0 O.D. cm$^{-1}$) was added 2.0 mL acetate buffer (10 mM, pH 4). After TA 20 μL and $H_2O_2$ 40 μL (0.2 M) were added, the resulting solution was incubated for 10 min and then centrifuged. The supernatant was measured with a fluorescence spectrophotometer (JASCO FP-8300).

### 4.6. Kinetic Analysis

To calculate the enzymatic parameters of different surface modified AuNPs, various concentrations of either $H_2O_2$ or TMB were prepared in acetate buffer at pH 4 (10 mM). According to the Michaelis–Menten Equation (1), the apparent kinetic parameters were calculated as below:

$$v = \frac{V_{\max}[S]}{K_m + [S]} \tag{1}$$

where $v$ represents the initial velocity, $[S]$ is the concentration of the substrate, $V_{\max}$ is the maximal reaction velocity, and $K_m$ is the Michaelis–Menten constant. Next, a plot of $1/[S]$ against $1/v$ based on the Lineweaver–Burk plot method (Equation (2)) resulted in a straight line. The corresponding $V_{\max}$ and $K_m$ could be obtained by the intercept and the slope of this line, respectively.

$$\frac{1}{v} = \frac{K_m}{V_{\max}[S]} + \frac{1}{V_{\max}} \tag{2}$$

## 5. Conclusions

In this study, we used the $S^{2-}$-induced inhibition of peroxidase-like activity to explore active sites of AuNPs. The peroxidase-like activities of AuNPs with various surface coatings could all be inhibited by $S^{2-}$ due to the interaction with Au(I) to form $Au_2S$. Accordingly, we considered that Au(I) might be

the active site of the peroxidase-like activity of AuNPs. XPS analysis showed that the Au(I) content was significantly reduced after the addition of $S^{2-}$ because the Au–S bond was more covalent than ionic in $Au_2S$. In addition, the $\Delta Au(I)\%$ without and with the addition of $S^{2-}$ could be recognized as the accessible percentage of Au(I) for the peroxidase-like reaction. The $\Delta Au(I)\%$ from large to small was unmodified AuNPs > GA-AuNPs > PVP-AuNPs > Cit-AuNPs > Cys-AuNPs. In the following, the peroxidase-like activities of all kinds of AuNPs were compared by the oxidation of peroxidase substrates and the production of ·OH, respectively. According to the kinetic analysis, the comparison of the $V_{max}$ values verified that the catalytic activity from high to low was as follows: unmodified AuNPs > GA-AuNPs > PVP-AuNPs > Cit-AuNPs > Cys-AuNPs. The $K_m$ values had no significant trend, implying that contributions from the electrostatic interactions between AuNPs and peroxidase substrates played only minor roles relative to that of the active sites. Importantly, the $\Delta Au(I)\%$ for different surface capped AuNPs were in good agreement with their corresponding peroxidase-like activities. Therefore, the accessible Au(I) on the surface was confirmed to be the main requisite for the peroxidase-like activity of AuNPs and the addition of $S^{2-}$ could assist to determine available active sites for different surface modified AuNPs. This work provides a new way to determine the surface accessibility mediated by different surface modification, which will guide the design of efficient AuNP-based peroxidase mimics.

**Supplementary Materials:** The following are available online at http://www.mdpi.com/2073-4344/9/6/517/s1. Figure S1: UV-Vis absorption spectra of all five kinds of AuNPs, Figure S2: Absorption spectra without and with the addition of $S^{2-}$ to examine the SPR band and the corresponding inhibition extent of the peroxidase-like activity for different AuNPs, Figure S3: Absorption spectra and TEM images of GA-AuNPs without and with the addition of $S^{2-}$, Figure S4: XPS spectra of Au 4f for all AuNPs without and with the addition of $S^{2-}$, Figure S5: XPS spectra of S2p for all AuNPs, Figure S6: Comparison of the peroxidase-like activities of different surface modified AuNPs, Figure S7–S11: The steady-state kinetics of all five kinds of AuNPs, Figure S12: Absorption spectra of Cit-AuNPs mixing with $H_2O_2$ in different incubation time.

**Author Contributions:** K.-C.C. and C.-F.S. performed experiments about peroxidase-like activities of AuNPs. C.-F.S. prepared catalyst samples and analyzed XPS data. P.-Y.Y. and P.-W.L. measured kinetic data and calculated kinetic parameters. K.-C.C. and C.-F.S. discussed results and wrote master's thesis in Chinese. C.-P.L. organized experimental results and wrote the manuscript.

**Funding:** This research was funded by the Ministry of Science and Technology of Taiwan, the grant number MOST-106-2113-M-030-005-MY2.

**Acknowledgments:** We thank the Ministry of Science and Technology of Taiwan (MOST-106-2113-M-030-005-MY2) for providing financial support for this research. Special thanks go to Swee Lan Cheah for her assistance on XPS measurements.

**Conflicts of Interest:** The authors declare no conflict of interest.

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
