# Peer review of "Revealing the Active Site of Gold Nanoparticles for the Peroxidase-Like Activity: The Determination of Surface Accessibility"

_catalysts, doi:10.3390/catal9060517_

Round 1
Reviewer 1 Report
The authors describe the synthesis and testing of peroxidase activity on gold nanoparticles stabilized with low and high molecular weight compounds. A method of evaluating the surface availability of nanoparticles was also proposed. The work is written very well. The results are presented correctly and very aesthetically. Actually, I have no comments on the article except one - the novelty of the research is doubtful. In the last 5 years, about 300 works on this topic have been published (Scopus). There are no such examinations and such materials as in the presented work but most of them touch a similar topic. Therefore, the novelty remains after reading this article.Despite this, the work, as it was written earlier, is very well presented and in my opinion, after minor editorial corrections (a few grammatical and stylistic errors, text from the template in the acknowledgments), it can be published as an article showing a certain methodology without, however, a large element of novelty.
Reviewer 2 Report
This manuscript is on the whole well written, clear and concise enough to avoid burdening. Unfortunately, it suffers from an 'original sin' wich in my opinion prevents its consideration for publication, at least in the present form. In fact, the Authors write about a "Peroxidase-like" activity of their nano-catalyst, and describe (correctly) its action mechanism as a Fenton-Like one. Action mechanism of peroxidases is quite not a Fenton-like one! The fact that a catalyst is capable of promote the oxidation of known peroxidase substrates by means of hydrogen peroxide cannot be taken as a proof of a peroxidase-like activity. Many recent articles deal with such a (false) peroxidase-like activity of several different catalysts, but such articles - most probably written by Authors poorly familiar with peroxidase enzymology - cannot make a justification to spread the misconception. incidentally, I have found the t the Authors cite (Ref. 11) an article which should deny the Fenton-like mechanism for the gold nanoparticles; inspection of that article shows that on the contrary it claim a Fenton-like action. In conclusion, the article should be profoundly revised according to these suggestions, starting from the very title...
Reviewer 3 Report
In this paper, the authors study the interaction of different capped Au NPs with S2- in order to possibly measure the amount of Au(I) on the surface and correlate it with their peroxidase-like activity. Although the work is technically impeccable and very well carried out, my concern with this work is it does not consider the fact that the reactivity and catalytic activity of nanoparticles is strongly dependent on their shape. Looking at the TEM images on Fig. 1, it can be see that these NPs are actually a mixture of different shapes and contain spherical, polygonal and prismatic particles. My question is thus if NPs with uniform but different shapes would behave similarly of not with respect to the access of sulfide. For this reason, I believe that the authors should prepare NPs with two or three different shapes and similar size and repeat their experiments in these conditions and, on the light of the results, they may be able to propose a mechanism for the peroxidase-like activity of Au NPs.
Reviewer 4 Report
The manuscript authored by Liu et. al presents research and discussion on the contribution of superficial Au(I) ions in the mechanism of peroxidase-like catalytic activity of gold nanoparticles. A key element of novelty in the presented research covers the use of selective blocking of Au(I) active sites due to the formation of covalent bonds with sulphide anions. Sulfur-containing compounds (mainly organic, such as thiols) are known inhibitors of nanocatalysts of oxidoreductase-like activity. This paper describes the attempt to use controlled blocking of the nanozyme surface (with the use of S2- chemisorption or steric stabilizers as surface ligands) as a method of studying the mechanism and the origin of intrinsic HRP-like activity of gold nanoparticles.
The provided idea of the use of sulphide ions is undoubtedly very promising. However, in my opinion, unambiguous confirmation of the postulated mechanism requires more extensive research. Therefore, I would encourage the authors to deepen the discussion based on the remarks below. The manuscript is well-written, with only few misspellings and language inaccuracies (see minor remarks). The main disadvantage of the manuscript is the section which covers the use of terephthalic acid for *OH determination and calculations of kinetic parameters according to Michaelis-Menten mechanism. These studies are quite weakly related to the main topic of the manuscript and their results were left without exhaustive discussion. In their present form, one can get the impression of unnecessary and routine characterization of known types of gold nanoparticles. In view of the above remarks, I can not recommend the acceptance of the manuscript in its present form and I encourage to revise it.
Most important remarks:
- The most widely accepted mechanism of Au-S bond formation (with the use of thiols) covers redox reaction between zerovalent gold and -SH moiety with the evolution of hydrogen (or reaction with thiolate in alkaline media). Such mechanism remains in contradiction with the thesis proposed by Authors. A more detailed discussion on the differences between the regioselectivity of chemisorption of S2- (in terms of pH/possible hydrolysis) and thiols would be valuable.
- Authors have discussed the influence of nanoparticle size on catalytic activity. Provided reference [13] reports the little influence of NPs size for the catalytic degradation of p-nitrophenol. There are several other references thoroughly addressing the relationship between AuNPs diameter and activity in reaction of organic dye reduction (Fenger et al., Phys. Chem. Chem. Phys. 2012, 14, 9343, Sau et al., J. Phys.Chem.B 2001, 105, 9266) or even HRP-like activity (Drozd et al., Nanotechnology, 2015, 26 (49): 495101). In these publications the origin of catalytic activity is explained on the basis of contribution of highly energetic, zerovalent gold atoms. The critical confrontation and discussion of both hypotheses would be also recommended.
- Page 4: The additional experiment with the use of lower concentration of sulphide ions would be required for more convincing confirmation of proposed mechanism.
- Fig. 4: Please provide better justification for provided determination of *OH radicals. In my opinion, these studies do not bring new information in context of the revealing of active sites of AuNPs.
- Page 7, Line 212: “Vmax values could reflect the catalytic activity (…)” – this statement is too general. Vmax (kcat) values are typically associated with kinetics/turnover number of catalyst rather than overall activity.
- Page 7, Line 219: Taken together, the addition (…) was the main requisite for the peroxidase-like activity of AuNPs.” – this conclusion is totally inappropriate in this place. There is a lack of justification and discussion regarding the determined kinetic parameters according to Michaelis-Menten model. The presentation of such results in the present form in the manuscript is in my opinion unreasonable.
- In general, the discussion section is quite brief and little detailed. I agree with Authors, that further, more careful studies (also with the use of already used methodologies) will be required to convincingly explore the HRP-like activity mechanism.
- What concentration of H2O2 was used for positive control with HRP? The use of 0.2 M H2O2 (the same as for AuNPs) is impossible due to enzyme inactivation.
- Page 9, Line 322: Please explain, why the sample was centrifuged before fluorescence measurement? (is the AuNPs-induced quenching a reason?)
- Page 10, Line 342: “In parallel, the peroxidase-like activities (…) were further confirmed by kinetic analysis” – this conclusion is in my opinion unjustified.
Minor remarks:
- Page 1, Line 34: „immunoassays” instead „immunoassay”
- Page 3, Line 121: please provide the accurate amount of added S2- ions (information on the level of inhibition is insufficient)
- Fig. 2, caption: the concentration of TMB (0.02 mM) is inconsistent with the value provided in the experimental section (Page 9, Line 302). Please explain this discrepancy.
- Page 5, Line 176: “unmodified” nanoparticles might be stabilized rather by borate (product of borohydride ion hydrolysis) than unstable BH4-.
- Page 8, Line 260: a word “sulphide” is missing
- Page 8, Line 295: How Cys-AuNPs were purified from the excess of reactants?
- Page 9, Line 314: “absorption spectra were collected at 420 nm (…) or at 652 nm” – please use the term “absorbance values” instead “spectra”.
Round 2
Reviewer 2 Report
The manuscript has been amended, and its quality has gone further forward. However, I maintain my criticism about the definition of '"peroxidase-like" which is in fact NOT a 'peroxidase-like' activity. The Authors' answer is rather captious, perhaps as to define "peroxidase-like" any catalytic activity using hydrogen peroxide as the oxidizing agent is undoubtedly trendy and could be imaginative and suggestive, allowing putative links to biotechnology, biomimicry, 'green' chemistry, and so on. Therefore, I maintain also my opinion about this manuscript
Author Response
Thanks for your comment. Since the name of the “peroxidase-like activity” has been widely used for nanomaterial-based artificial enzymes (ref. 1-4), we followed the research in this field and then used the same name. In addition, our previous reply has explained that our experiments excluded the Fenton-like mechanism for AuNPs (consistent with ref. 11). Moreover, the enzymatic examination and kinetics were used in our work. Therefore, using the peroxidase-like activity of AuNPs was the appropriate choice.

Reviewer 3 Report
The authors have addressed my previous comments and the manuscript can now be accepted in its present form.
Author Response
We thank the reviewer for their valuable comments.

Reviewer 4 Report
The introduced changes went in a right direction, however more of the important issues clarified in a response to reviewers should be also included in a body of a manuscript. Moreover, the language used requires polishing (especialy the revised part).
Author Response
We thank the reviewer for their valuable comments. These comments are very constructive and will help us to improve the manuscript. In fact, we kept some response only for reviewer because further experiments are still ongoing. For example, 2-MBI experiments showed the inhibition of the peroxidase-like activity, but the adsorption amount of 2-MBI failed to correlate with the peroxidase-like activity. This is because the addition of 2-MBI induced severe aggregation of colloidal AuNPs. Although I considered that the -SH group maybe pick out accessible Au(0) on the surface of AuNPs for the peroxidase-like activity, we have no direct evidence yet in this work. Therefore, the 2-MBI data was removed and temporary explanation was addressed (page7-8, Line 242-252) in the revised manuscript. Finally, the article has been completed in the revised version.
